

# Functional properties of bacterial communities in water and sediment of the eutrophic river-lake system of Poyang Lake, China

Ze Ren[1,2], Xiaodong Qu[1,3], Wenqi Peng[1,3], Yang Yu[1,3] and Min Zhang[1,3]

[1] State Key Laboratory of Simulation and Regulation of Water Cycle in River Basin, China Institute of Water Resources and Hydropower Research, Beijing, China
[2] Flathead Lake Biological Station, University of Montana, Polson, MT, USA
[3] Department of Water Environment, China Institute of Water Resources and Hydropower Research, Beijing, China

Corresponding author
Xiaodong Qu,
quxiaodong@iwhr.com

## ABSTRACT

In river-lake systems, sediment and water column are two distinct habitats harboring different bacterial communities which play a crucial role in biogeochemical processes. In this study, we employed Phylogenetic Investigation of Communities by Reconstruction of Unobserved States to assess the potential functions and functional redundancy of the bacterial communities in sediment and water in a eutrophic river-lake ecosystem, Poyang Lake in China. Bacterial communities in sediment and water had distinct potential functions of carbon, nitrogen, and sulfur metabolisms as well as phosphorus cycle, while the differences between rivers and the lake were inconspicuous. Bacterial communities in sediment had a higher relative abundance of genes associated with carbohydrate metabolism, carbon fixation pathways in prokaryotes, methane metabolism, anammox, nitrogen fixation, and dissimilatory sulfate reduction than that of water column. Bacterial communities in water column were higher in lipid metabolism, assimilatory nitrate reduction, dissimilatory nitrate reduction, phosphonate degradation, and assimilatory sulfate reduction than that of sediment bacterial communities. Furthermore, the variations in functional composition were closely associated to the variations in taxonomic composition in both habitats. In general, the bacterial communities in water column had a lower functional redundancy than in sediment. Moreover, comparing to the overall functions, bacterial communities had a lower functional redundancy of nitrogen metabolism and phosphorus cycle in water column and lower functional redundancy of nitrogen metabolism in sediment. Distance-based redundancy analysis and mantel test revealed close correlations between nutrient factors and functional compositions. The results suggested that bacterial communities in this eutrophic river-lake system of Poyang Lake were vulnerable to nutrient perturbations, especially the bacterial communities in water column. The results enriched our understanding of the bacterial communities and major biogeochemical processes in the eutrophic river-lake ecosystems.

## INTRODUCTION

Lakes and their tributaries are highly linked ecosystems in multiple ways, especially through materials transported from the watershed to the lake through river systems (*Cole et al., 2006*; *Marcarelli & Wurtsbaugh, 2009*; *Jones, 2010*; *Ylla et al., 2013*). Microbial communities in lake and its tributaries have different taxonomic compositions (*Ren et al., 2017a*, *2019*). In lake ecosystems, water and sediment are two distinct realms and interact closely through biogeochemical processes (*Parker et al., 2016*). These two habitats host tremendous diversity of microorganisms (*Lozupone & Knight, 2007*; *Röeske et al., 2012*; *Huang et al., 2016*), which constitute distinct microbial communities in sediment and water column (*Briée, Moreira & López-García, 2007*; *Nishihama et al., 2008*; *Ren et al., 2019*). However, the functional differences of bacterial communities in sediment and water column of lake-river systems were not well studied.

In aquatic ecosystems, bacterial communities play an extremely important role in transformation, accumulation, and migration of nutrients and other elements, as well as in energy conversion and material recycling (*Cotner & Biddanda, 2002*; *Van Der Heijden, Bardgett & Van Straalen, 2008*; *Newton et al., 2011*). Bacterial communities exhibit high compositional and functional variability (*Newton et al., 2011*). Functional traits are valuable ecological markers to understand the bacterial community assembly (*Barberan et al., 2012*). Moreover, microbial metabolic activities can influence water quality through the storage and release of nutrients (*Nielsen et al., 2006*; *Hupfer & Lewandowski, 2008*). Thus, it is crucial to understand the roles of bacterial communities in biogeochemical cycling and elucidate their responses to environmental changes by unraveling their functional potentials (*Green, Bohannan & Whitaker, 2008*; *Fierer et al., 2012*; *Freedman & Zak, 2015*; *Ren et al., 2017b*). In addition, previous studies suggested that distinct taxa can share specific functional attributes while closely related taxa may exhibit distinct functional features (*Allison & Martiny, 2008*; *Philippot et al., 2010*; *Fierer et al., 2012*; *Dopheide et al., 2015*). Thus, the relationships between taxonomic and functional differences can help to elucidate functional redundancy and stability of bacterial communities.

Changes in water quality and sediment properties drive the variation of bacterial communities which regulate the core biogeochemical processes such as carbon and nitrogen metabolisms in aquatic ecosystems (*Liu et al., 2018*; *Wang et al., 2018*; *Yao et al., 2018*). As the largest freshwater lake in China, Poyang Lake is fed by five tributaries and is experiencing aggravated nutrient loading from agriculture and urbanization of the catchment in recent decades (*Wang & Liang, 2015*; *Liu, Fang & Sun, 2016*). The increase in nutrient inputs caused by agriculture, urbanization, and industry has significantly degraded water quality and ecological integrity of Poyang Lake with serious eutrophication (*Wang et al., 2015*; *Zhang et al., 2015*; *Liu, Fang & Sun, 2016*). In the river-lake systems of Poyang Lake, our previous study has shown that the taxonomic composition of bacterial communities in lake sediment (SL), river sediment (SR), lake water (WL), and

river water (WR) had distinct spatial distribution patterns and close relationships with nutrients (*Ren et al., 2019*). However, our understanding of the functions mediated by the bacterial communities in the sediment and water of this linked river-lake ecosystem is still limited. To reveal the functional potentials of bacteria, metagenomic sequencing has been used in a growing number of studies (*Mackelprang et al., 2011*; *Fierer et al., 2012*; *Llorens-Marès et al., 2015*). Alternatively, Phylogenetic Investigation of Communities by Reconstruction of Unobserved States (PICRUSt) is cheaper, faster, and reliable (*Wilkinson et al., 2018*) and has been widely used to infer the functional profile of the bacterial communities using 16S rRNA genes and a reference genome database to predict the functional composition of a metagenome (*Langille et al., 2013*). In this study, we predicted metagenomes from 16S rRNA gene sequences and classified into Kyoto Encyclopedia of Genes and Genomes (KEGG) Orthologs (KOs) using PICRUSt. The KOs associated with carbon, nitrogen, and sulfur metabolisms as well as phosphorus cycle were identified from KEGG database (*Kanehisa & Goto, 2000*). We aimed to reveal the functional properties of bacterial communities in SL, SR, WL, and WR in the river-lake system of Poyang Lake, including (1) metabolism pathways of major functions, (2) influences of nutrient variables on functional compositions, and (3) functional redundancy.

## MATERIALS AND METHODS

### Study area and field sampling

Poyang Lake is located in the lower reach of Yangtze River. With a surface area over 4,000 km$^2$ (in summer), it is the largest freshwater lake in China. There are five rivers (Fuhe, Ganjiang, Xinjiang, Raohe, and Xiushui) feeding Poyang Lake and one outlet connecting to Yangtze River (Fig. 1). The annual runoff of Poyang Lake is 152.5 billion m$^3$, accounting for 16.3% annual runoff of Yangtze River. Poyang Lake is a shallow seasonal lake and a typical water-carrying and throughput lake restricted by the water level of Yangtze River and the inflows of the five tributaries (*Fang et al., 2011*; *Zhao et al., 2011*). The high and low water levels of Poyang Lake are 20.69 and 9.82 m above the sea level, respectively (*Liao, Yu & Guo, 2017*). The average water depth is 8.4 m (*Wang & Liang, 2015*). Poyang Lake has been suffering persistent eutrophication (*Liao, Yu & Guo, 2017*). Previous research shown that cyanobacteria blooms have been observed in Poyang Lake since 2000 (*Liu et al., 2016*) but only occur periodically and regionally (*Liu & Fang, 2017*). We did not find cyanobacteria bloom during our sampling in early August 2017.

We collected samples from Poyang Lake and its tributaries in 10 and 24 sample sites, respectively (Fig. 1). In each sample site, a handheld meter (YSI Professional Plus, Yellow Springs, OH, USA) was used to measure water temperature (Temp), dissolved oxygen (DO), pH, and conductivity (Cond) in situ. Secchi disk depth was measured as well. Water samples were collected at the depth of 0.5 m using a Van Dorn water sampler. A total of 200 mL water was filtered onto a 0.2-μm Polycarbonate Membrane Filter (Whatman, UK), which was immediately frozen in liquid nitrogen in the field and stored at −80 °C in the lab until DNA extraction. Another 500 mL water was acid fixed in the field and transported to the laboratory at 4 °C for chemical analyses. Sediment samples were collected using a Ponar Grab sampler at the depth of 5.5–6.5 m in Poyang Lake and of

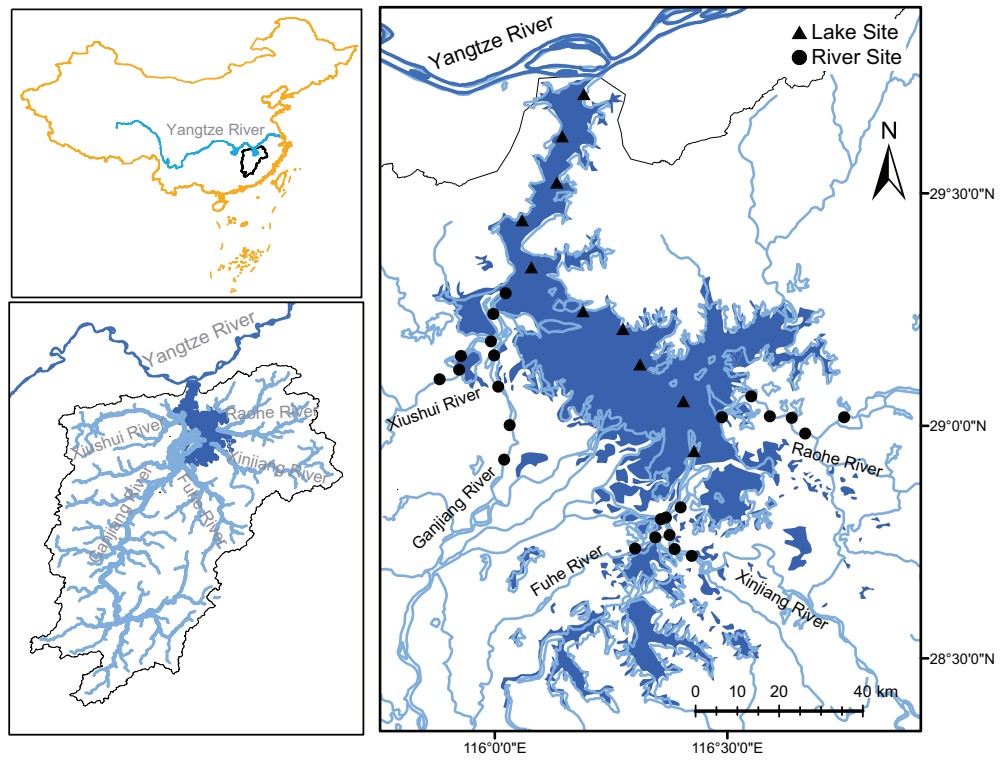

**Figure 1 Study area and sampling sites.** Samples were collected from the surface water and sediment of Poyang lake and its fiver tributaries (Xiushui, Ganjiang, Fuhe, Xinjiang, and Raohe). This figure was modified from *Ren et al. (2019)*.

3.9–5.8 m in the tributaries. The top five-cm sediment was homogenized by stirring with a spatula, collected in a sterile centrifuge tube, and immediately frozen in liquid nitrogen in the field for DNA extraction. The remaining sediment was collected in a clean Ziploc bag for chemical analyses.

For water samples, total nitrogen (TN), nitrate ($NO_3^-$), ammonium ($NH_4^+$), total phosphorus (TP), and soluble reactive phosphorus (SRP) were analyzed according to the Clean Water Act Analytical Methods (*United States Environmental Protection Agency (EPA), 2017*). DOC was analyzed using a TOC Analyzer (TOC-VCPH; Shimadzu Scientific Instruments, Kyoto, Japan). Detailed information of water sample analyses was provided in our previous study (*Ren et al., 2019*). For sediment samples, TN was analyzed using the modified Kjeldahl method (HJ717-2014). $NO_3^-$ and $NH_4^+$ were analyzed using UV spectrophotometry method (HJ634-2012). TP was analyzed using alkali fusion-Mo-Sb Anti spectrophotometric method (HJ632-2011). Total organic carbon (OC) was analyzed using Potassium dichromate oxidation spectrophotometric method (HJ615-2011). Organic nitrogen was analyzed using acid hydrolysis method (*Bremner, 1965*). Organic phosphorus was analyzed using SMT method (*Ruban et al., 1999*).

## DNA extraction, PCR, and sequencing

DNA was extracted from the filter and sediment (0.5 g) samples using the TIANGEN-DP336 soil DNA Kit (TIANGEN-Biotech, Beijing, China) following manufacturer protocols.

Extracted DNA samples were quantified using a Qubit 2.0 Fluorometer (Invitrogen, Carlsbad, CA, USA). The V3 and V4 regions were amplified using the forward primer 347F 5′-CCTACGGRRBGCASCAGKVRVGAAT-3′ and the reverse primer 802R 5′-GGACTACNVGGGTWTCTAATCC-3′ (GENEWIZ, Inc., South Plainfield, NJ, USA) (*Ren et al., 2019*). PCR was performed using the following program: initial denaturation at 94 °C for 3 min, 24 cycles of denaturation at 94 °C for 30 s followed by annealing at 57 °C for 90 s and extension at 72 °C for 10 s, and final extension step at 72 °C for 10 min. Amplified DNA was verified by electrophoresis of PCR mixtures in 1.0% agarose in 1× TAE buffer and purified using the Gel Extraction Kit (Qiagen, Hilden, Germany). DNA libraries were validated by Agilent 2100 Bioanalyzer (Agilent Technologies, Palo Alto, CA, USA), and quantified by Qubit 2.0 Fluorometer (Invitrogen, Carlsbad, CA, USA). DNA libraries were multiplexed and loaded on an Illumina MiSeq instrument (Illumina, San Diego, CA, USA) according to manufacturer's instructions.

## Sequence analysis and functional gene prediction

Raw sequence data was processed using the software package QIIME 1.9.1 (*Caporaso et al., 2010*). The forward and reverse reads were joined and assigned to samples based on barcode and truncated by cutting off the barcode and primer sequence. Then the sequences were quality filtered, and the chimeric sequences were removed. Sequences which did not fulfill the following criteria were discarded: sequence length <200 bp, no ambiguous bases, mean quality score ≥20 (*Ren et al., 2019*). The effective sequences were grouped into operational taxonomic units (OTUs) at 97% sequence identity level against the Greengenes 13.8 database (*McDonald et al., 2012*). Then the functional potentials of the bacterial communities were predicted using PICRUSt 1.1.0 and the nearest sequence taxon index (NSTI) was calculated to indicate the accuracy of PICRUSt prediction (*Langille et al., 2013*). The average NSTI was 0.146, indicating high accuracy (*Langille et al., 2013*). Then, the predicted metagenomes were further classified into KEGG KOs. The KOs associated with carbon, nitrogen, and sulfur metabolism as well as phosphorus cycle were identified from KEGG database (*Kanehisa & Goto, 2000*; *Bergkemper et al., 2016*). The Raw sequence data are available at National Center for Biotechnology Information (PRJNA436872, SRP133903).

## Statistical analysis

To reveal the functional differences (overall metagenomic functions and the major functions, including carbon metabolism, nitrogen metabolism, phosphorus cycle, and sulfur metabolism) between the bacterial communities in different habitats of the river-lake ecosystem of Poyang Lake, non-metric multidimensional scaling (NMDS) and analysis of variance using distance matrices (ADONIS) were applied using the Vegan package 2.4–6 (*Oksanen et al., 2007*) based on the relative abundance of KOs. Differences of the major pathways associated to carbon metabolism, nitrogen metabolism, phosphorus cycle, and sulfur metabolism between the bacterial communities in sediment and water column were tested using analysis of variance and the *P*-values were adjusted by FDR correction. Linear regression was used to assess the relationships between

taxonomic and functional dissimilarities, revealing functional redundancy of the bacterial communities (stronger linear regression indicates lower functional redundancy) (*Yang et al., 2017*; *Galand et al., 2018*). Taxonomic and functional dissimilarities were calculated as Bray–Curtis distances based on the phylogenetic and metagenomic compositions (relative abundance of OTUs and KOs, respectively). The differences of linear regression slopes were compared using analysis of covariance (ANCOVA). Distance-based redundancy analysis (dbRDA) was conducted using Vegan package to reveal the relationships between environmental variables (normalized using "normalize" method) and overall functional compositions (relative abundance of KOs, Hellinger transferred) of bacterial communities in sediment and water column, and the significance of the nutrient variables was tested using Envfit function in R. Mantel tests were applied to assess the relationships between nutrient factors and major functions and the *P*-values were adjusted by FDR correction. All the analyses were conducted in R 3.4.4 (*R Core Team, 2017*).

## RESULTS

### Functional differences

In total, 6,295 and 6,187 KOs were detected in bacterial communities in sediment and water column. Profound differences were detected in functional compositions between sediment and water. NMDS and ADONIS showed that bacterial communities in LS and RS were significantly ($P < 0.05$) different to LW and RW (LS vs. LW and RS vs. RW), respectively (Fig. 2). However, there was no difference between Poyang Lake and its tributaries (LS vs. RS and LW vs. RW, Fig. 2). For carbon metabolism, we detected 242 KOs associated to central carbon metabolism pathways (ko01200) based on the KEGG database. Carbohydrate metabolism, carbon fixation pathways in prokaryotes, and methane metabolism had a higher relative abundance of associated genes in the bacterial communities in sediment than in water (Fig. 3A). However, the lipid metabolism had a higher relative abundance in water than in sediment (Fig. 3A). For the nitrogen metabolism, we detected 41 KOs associated to nitrogen metabolism pathways (ko00910 in KEGG database). Bacterial communities had a higher relative abundance of genes associated to anammox and nitrogen fixation in sediment than in water (Fig. 3B). However, assimilatory nitrate reduction to ammonia (ANRA) and dissimilatory nitrate reduction to ammonia (DNRA) had a higher relative abundance in water than in sediment (Fig. 3B). For phosphorus, we detected 43 KOs associated to phosphorus cycle. Phosphonate degradation had a higher relative abundance in water than in sediment (Fig. 3C). For sulfur metabolism, we detected 45 KOs associated to the sulfur metabolism pathways (ko00920 in the KEGG database). Assimilatory sulfate reduction had a lower relative abundance while dissimilatory sulfate reduction had a higher relative abundance in sediment than in water (Fig. 3D).

### Environmental influences

The results of dbRDA indicated that the overall functional compositions of bacterial communities in sediment were significantly correlated with TP and $NO_3^-$ (Fig. 4A). The first two axes explained 31.85% of the functional variation (dbRDA 1: 19.21%; dbRDA 2: 12.64%).

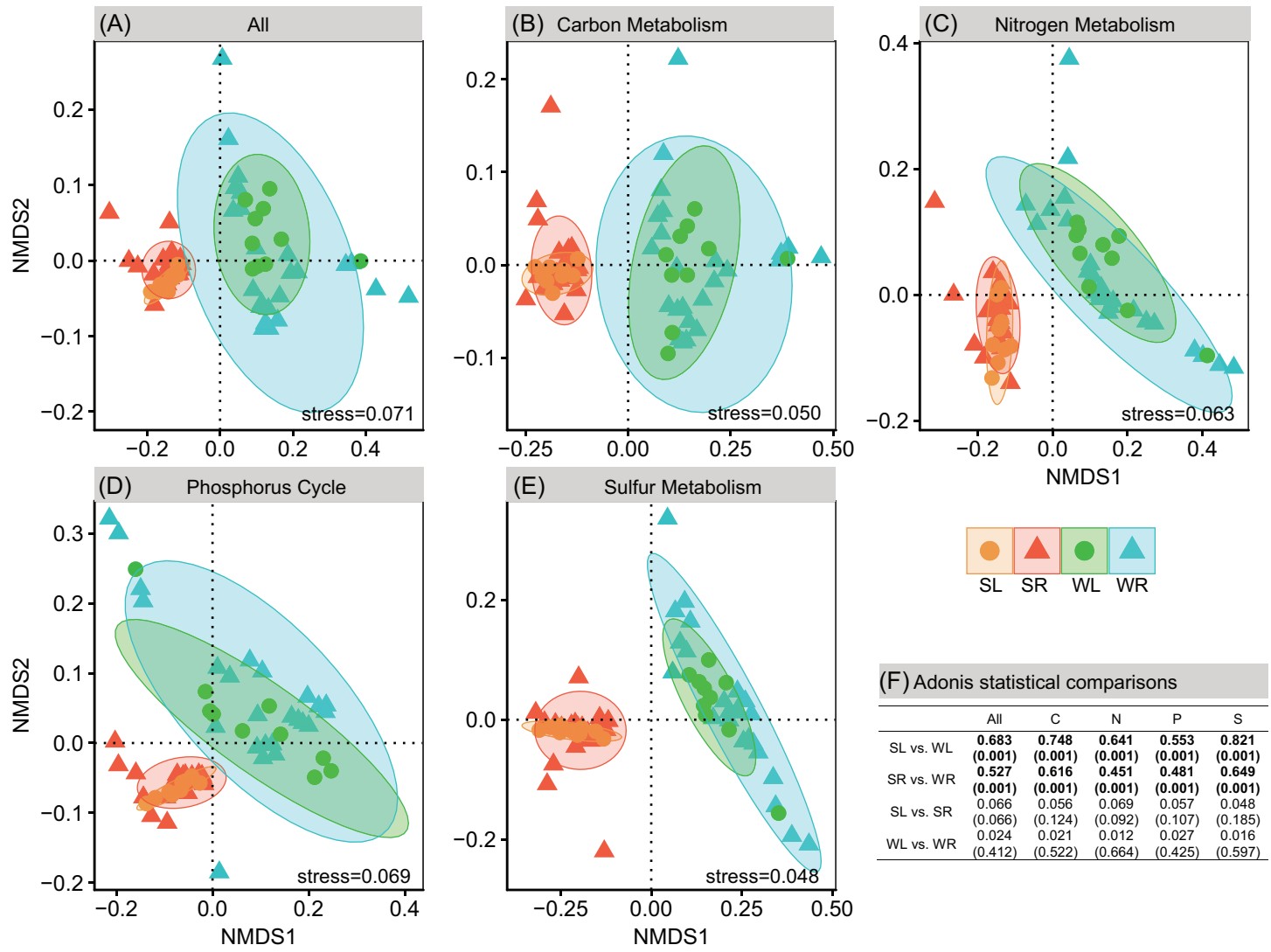

**Figure 2 Functional differences between habitats.** (A–E) Non-metric multidimensional scaling analysis of potential functions composition in terms of overall functions, carbon metabolism, nitrogen metabolism, phosphorus cycle, and sulfur metabolism. (F) Pairwise dissimilarity tests of functional composition between different habitats using ADONIS. The numbers outside the bracket are "$R^2$." $P$-values are in bracket.

For the bacterial communities in water column, the overall functional compositions were significantly correlated with TN, $NO_3^-$, TP, TN:TP, DOC:DIN, DIN:SRP, as well as Cond, Temp, DO, and pH (Fig. 4B). The first two axes explained 67.03% of the functional variation (dbRDA 1: 47.79%; dbRDA 2: 19.24%). Mantel tests further demonstrated that the spatial variations of the major biogeochemical processes (C-metabolism, N-metabolism, P-cycle, and S-metabolism) were significantly influenced by TP and $NO_3^-$ in sediment, and by TN, TP, SRP, DOC:DIN, and DOC:SRP in water column (Fig. 5).

## Functional redundancy

Linear regressions between taxonomic and functional dissimilarities showed that the variations in metagenomic functional composition (overall, C-metabolism, N-metabolism,

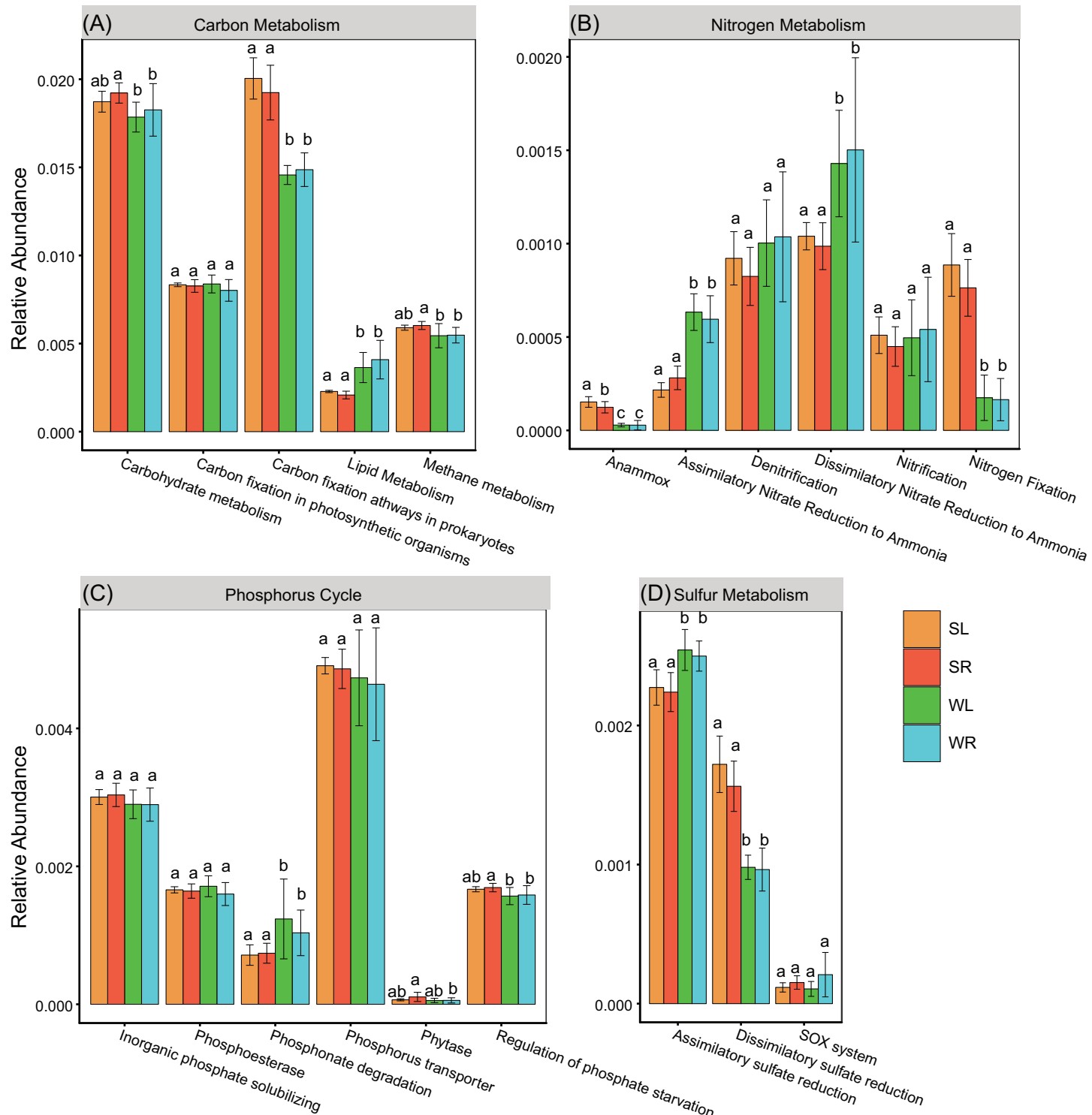

**Figure 3 Relative abundance of genes associated to major pathways in (A) Central carbon metabolism, (B) nitrogen metabolism, (C) phosphorus cycle, and (D) sulfur metabolism.** For each pathway, the same lowercase letter indicates a non-significant difference, whereas the different letter indicates a significant difference between habitats (ANOVA, $P < 0.05$). $P$-values were adjusted by FDR correction.

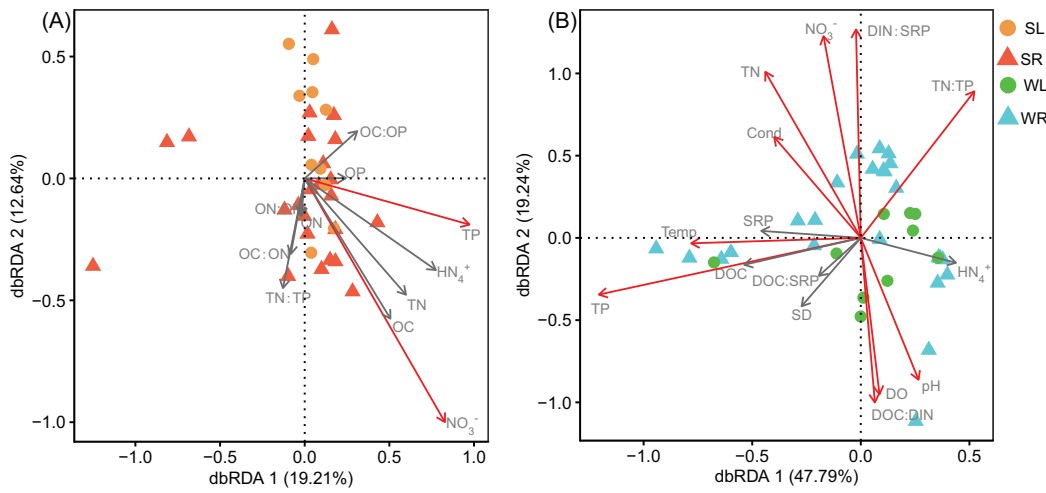

**Figure 4 Biplot of distance-based redundancy analyses (dbRDA) showing the relationship between functional composition and nutrient variables in (A) sediment and (B) water.** The red arrows represent the significant variables (envfit, $P < 0.05$).

**Figure 5 Mantel tests between major functions and nutrient variables of (A) sediment and (B) water based on Spearman correlation.** Significant correlations ($P < 0.05$) were colored. $P$-values were adjusted by FDR correction.

P-cycle, and S-metabolism) were closely associated with the variations in phylogenetic composition (Fig. 6). However, sediment bacterial communities had significantly smaller slopes than bacterial communities in water column (ANCOVA, $P < 0.05$, Fig. 6F). For the major functions in sediment, nitrogen metabolism had a higher slope, followed by sulfur metabolism, carbon metabolism, and phosphorus cycle (Fig. 6). In the water column, however, nitrogen metabolism and phosphorus cycle had higher slopes than carbon and sulfur metabolisms (Fig. 6). The results suggested that bacterial communities in sediment had higher functional redundancy than in the water column. Moreover, bacterial communities had lowest functional redundancy for nitrogen metabolism but highest functional redundancy for phosphorus cycle in sediment, while had lowest redundancy for both nitrogen metabolism and phosphorus cycle in the water column.

## DISCUSSION

In this study, the functional composition of bacterial communities in the river-lake system of Poyang Lake were different between water and sediment (LS vs. LW and RS vs. RW),

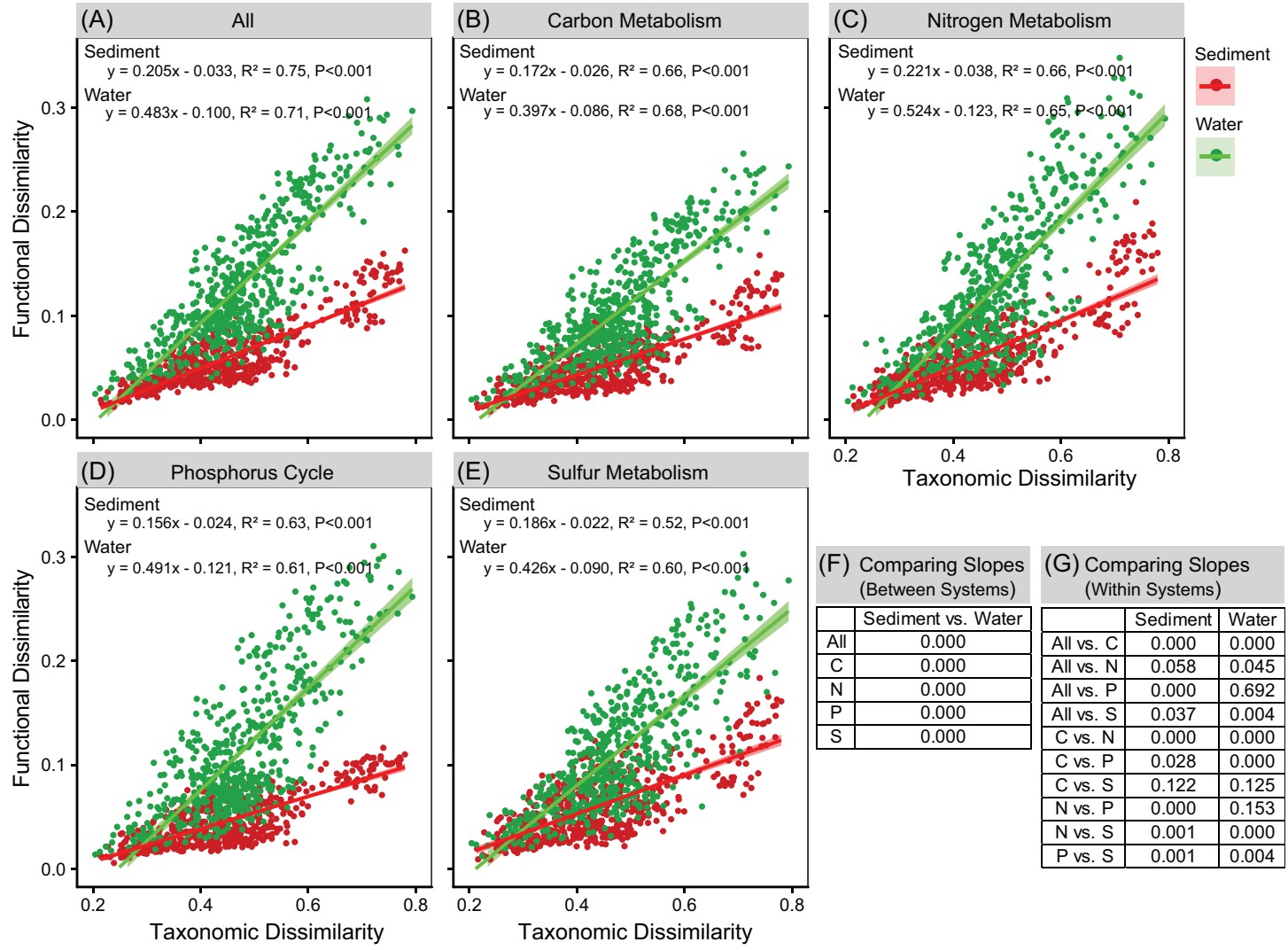

**Figure 6 Linear regressions between taxonomic and functional dissimilarities (A–E).** One point represents one sample pair. Shadow area denotes the 95% confidence interval. (F–G) Statistical test of the linear regression slopes between systems and within systems using ANCOVA.

while no different between tributaries and the lake itself (LS vs. RS and LW vs. RW). In our previous study of the river-lake system of Poyang Lake (*Ren et al., 2019*), bacterial communities were taxonomically different between sediment and water. It has been well demonstrated that sediment and water had distinct bacterial communities (*Jiang et al., 2006*; *Nishihama et al., 2008*; *Lu et al., 2016*), which might determine significant functional differences (*Fierer et al., 2012*; *Ren et al., 2017a*). However, the taxonomical differences of bacterial communities between Poyang Lake and its tributaries were significant but smaller compared to the differences between sediment and water (*Ren et al., 2019*). In generally, bacterial communities were more taxonomically different than functional different (*Louca et al., 2017*; *Ren et al., 2017a*). Thus, the small differences in taxonomic composition of bacterial communities did not lead to their functional differences between Poyang Lake and its tributaries.

This study showed that carbohydrate metabolism, carbon fixation pathways in prokaryotes, and methane metabolism had a higher relative abundance in the bacterial communities in sediment than in water, and the lipid metabolism had a higher relative abundance in water than in sediment. The results suggested that bacterial communities in sediment and water had distinct carbon metabolism pathways. Organic matter (OM) transported by river provides fueling aquatic food webs as a major source of energy and is also a significant component of the global carbon cycle (*Cole et al., 2007*; *Battin et al., 2008*; *Smith & Kaushal, 2015*). In freshwater ecosystems, OM is a heterogeneous mixture including allochthonous materials contributed by soil and plant litter inputs from terrestrial ecosystems and autochthonous materials contributed by primary producers in freshwater ecosystems (*Webster & Meyer, 1997*). OM is consisted of carbohydrates, proteins, lipids, lignins, and other compounds in aquatic ecosystems (*Thurman, 2012*). Microorganisms are key biogeochemical agents in the generation, transformation, and mineralization of OM (*Horvath, 1972*). Variations of OM in its source and composition, as well as the bioavailability of its components determine the spatial patterns of bacterial composition and functional diversity (*Hoostal & Bouzat, 2008*; *Wang et al., 2018*). In aquatic ecosystems, sediment and water column have distinct redox environments (*Röeske et al., 2012*), and the OM derives from different sources with different compositions (*Hedges, Clark & Come, 1988*). These differences might lead to the distinct carbon metabolisms between water and sediment. For example, the reduction condition in sediment is benefit to methane production (*Koyama, 1963*; *He et al., 2015*; *Liu & Xu, 2016*). In sediment, methane-oxidizing and sulfate-reducing bacteria also play the roles in carbon fixation (*Kellermann et al., 2012*).

Our study also showed that sediment and water column were significantly different in nitrogen metabolism, suggesting different nitrogen use strategies. In the past century, the nitrogen entering freshwater ecosystem has been increased more than twofold by anthropogenic activities (*Schlesinger, 2009*; *Meunier et al., 2016*), contributing to eutrophication in lake and coastal ecosystems (*Nixon, 1995*; *Smith, 2003*). Poyang Lake has been facing serious threat of eutrophication (*Wang et al., 2015*; *Zhang et al., 2015*; *Liu, Fang & Sun, 2016*) because of the aggravated nutrient loading from agriculture and urbanization of the catchment in recent decades (*Wang & Liang, 2015*; *Liu, Fang & Sun, 2016*). Nitrogen has many different chemical forms from the oxidation state of nitrate (+5) to the reduction state of ammonia (−3) and is cycled by a suite of biogeochemical processes (*Ollivier et al., 2011*), including four reduction pathways (denitrification, nitrogen fixation, ANRA, and DNRA) and two oxidation pathway (anammox and nitrification) (*Lamba et al., 2017*). In aquatic ecosystems, denitrification is the main biological process turning nitrate to dinitrogen and nitrous oxide (*Tiedje et al., 1983*; *Seitzinger, 1988*) and anammox is another important pathway turning nitrite and ammonia to dinitrogen (*Dalsgaard et al., 2003*; *Kuypers et al., 2003*). Both denitrification and anammox play important roles in removing nitrogen from aquatic ecosystems. In our study, bacterial communities in both sediment and water had a high relative abundance of the genes associated to denitrification, suggesting strong potentials in nitrogen removal. Many previous studies have demonstrated that rivers and lakes are hot spots to remove N inputs

to surface waters from terrestrial environments (*Wollheim et al., 2008*; *Harrison et al., 2009*; *Beaulieu et al., 2011*). Denitrification can be limited by the supply of $NO_3^-$ and OC, as well as redox potential (*Van Kessel, 1977*; *Seitzinger, 1988*). Furthermore, nitrification is also an important process in the N cycle and couples with denitrification (*Jenkins & Kemp, 1984*; *Nils, 2003*), especially in the shallow lakes. In the eutrophic river-lake system of Poyang Lake, the high contents of OM and $NO_3^-$ in water and sediment can facilitate denitrification. Moreover, the respiration in sediment can provide an anoxic environment and promote sediment denitrification. On the other hand, it has also been supported by many studies that aerobic denitrification can be performed by a broad range of bacterial organisms under an aerobic environment (*Ji et al., 2015*; *Lv et al., 2017*). In our study, the high potential denitrification (potential $NO_3^-$ reductions) in water might be performed through aerobic denitrification with the facilitation of high supplement of OM and $NO_3^-$. In addition to denitrification, bacterial communities in sediment had a significantly higher relative abundance of the genes associated to anammox than in water. It was found that anammox can coupled to nitrate reduction to contribute substantially to produce dinitrogen in sediments (*Thamdrup & Dalsgaard, 2002*). These results suggested that the bacterial communities in water and sediment of this eutrophic river-lake system had strong functional potentials but different strategies in nitrogen removal. In contrast to nitrogen removal, bacterial communities in sediment also had a higher relative abundance of genes associated with nitrogen fixation. In fact, the genetic potential of nitrogen fixation is pervasive among the domains of Bacteria and Archaea (*Zehr et al., 2003*). Nitrogen fixation and denitrification can co-occur in sediments through heterotrophic nitrogen fixation (*Newell et al., 2016*). We have underestimated the importance of heterotrophic sediment nitrogen fixation in the past, which can be an important source of nitrogen even under higher inorganic nitrogen concentrations (*Fulweiler & Heiss, 2014*; *Newell et al., 2016*). Examining the expression of the genes encoding for nitrogenase (such as nifD, nifH, nifK, and anfG) in the bacterial communities can help us understand the nitrogen fixation potential in freshwater ecosystems. In our study, the high relative abundance of genes associated to nitrogen fixation suggested a significant nitrogen fixation potential in sediment in Poyang Lake and its tributaries. In nitrogen metabolism pathways, both ANRA and DNRA had a higher relative abundance in water than in sediment, suggesting strong potentials of nitrate reduction to ammonia for bacterial communities in water column. ANRA and DNRA serve distinct cellular functions (*Lamba et al., 2017*): ANRA consumes energy and provides ammonium for cell to synthesize amino acids and nucleotides, while NDRA generates ATP in absence of oxygen and retains the nitrogen in the form of $NH_4^+$ for further biological processes (*Zumft, 1997*).

Phosphorus is an essential element in all ecosystems used by all living organisms. Bacteria plays a pivotal role in natural phosphorus cycles on the earth (*Ohtake et al., 1996*; *Kononova & Nesmeyanova, 2002*). In this study, the results showed that phosphonate degradation had a higher relative abundance in water than in sediment. Phosphonates are characterized by direct carbon-to-phosphorus bonds, which are resistant to chemical hydrolysis and thermal degradation (*Ohtake et al., 1996*; *Kononova & Nesmeyanova, 2002*).

In polluted freshwater ecosystems, large quantities of phosphonates are xenobiotics, such as pesticides, antibiotics, and detergent additives (*Schowanek & Verstraete, 1990*). It has been revealed that phosphonates are significantly removed from marine basin due to rapid release and remineralization (*Benitez-Nelson et al., 2004*). Our study suggested that bacterial communities in water column are important in phosphonates removal.

For sulfur metabolism in the eutrophic river-lake system of Poyang Lake, our study showed that the assimilatory reduction is more common than dissimilatory reduction and the bacterial communities in water had a higher assimilatory sulfur reduction potential while lower dissimilatory sulfate reduction than in sediment. Sulfur is an important element required for some cellular components related to proteins. In the sulfur metabolism of bacteria, assimilatory sulfate reduction commences with the incorporation of sulfide radical for the biosynthetic cycle. Thus, during assimilatory sulfate reduction, there is no sulfide produced. For some microorganisms, sulfur compounds are utilized in dissimilatory and energy-yielding metabolic processes, which takes place in anaerobic respiration. During dissimilatory sulfate reduction, sulfate ion is used as the terminal electron acceptor and is reduced to produce sulfide, in the meantime, OC is mineralized with producing of carbon dioxide. In SLs, dissimilatory sulfate reduction can account for a significant fraction of OC mineralization, especially in eutrophic lakes with high availabilities of OM and sulfate (*Holmer & Storkholm, 2001*). The differences of sulfur metabolism between water and sediment shed light on sulfur use strategies of bacterial communities in these two distinct habitats.

As discussed above, biogeochemical cycles of C, N, P, and S are important ecological functions in freshwater ecosystems. In our study, mantel tests showed significant correlations between taxonomic and functional dissimilarity matrixes (beta diversities), suggesting that the overall changes in potential functions, as well as the changes of potential metabolisms of C, N, P, and S were closely associated with changes in taxonomic compositions of the bacterial communities. Functional redundancy always exists in natural ecosystems (*Cardinale, Nelson & Palmer, 2000*; *Rosenfeld, 2002*; *Allison & Martiny, 2008*) and is measured by the correlation between taxonomic and functional gene diversities (*Fierer et al., 2013*; *Yang et al., 2017*). Functional redundancy occurs when different organisms execute a similar function, remaining functional stabilization of communities upon species loss (*Rosenfeld, 2002*; *Nyström, 2006*). Our results showed that the bacterial communities in water column had a lower redundancy of overall functions than in sediment. Moreover, compared to overall functions, sediment bacterial communities had lower functional redundancy of N metabolism, and bacterial communities in water column had lower functional redundancy of N metabolism and P cycle. In bacterial communities, functional redundancy is expected to allow bacterial communities to have a certain extent of resistance and resilience in facing environmental perturbations (*Allison & Martiny, 2008*; *Bowen et al., 2011*). The results suggested that the bacterial communities in water column were less stable than bacterial communities in sediment. Moreover, N metabolism and P cycle was more vulnerable to environmental perturbations than C and S metabolisms, influencing nutrient biogeochemical processes in the eutrophic river-lake system of Poyang Lake.
## CONCLUSIONS

In this study, we assessed the functional properties of bacterial communities in SL, SR, WL, and WR in the river-lake system of Poyang Lake. In general, the results showed that bacterial communities in sediment and water had distinct potential functions in the biogeochemical processes of carbon, nitrogen, phosphorus, and sulfur. However, there was no difference between tributaries and the lake itself. Moreover, bacterial communities in water column had a lower functional redundancy than in sediment. Comparing to the overall functions within systems, bacterial communities had lower functional redundancy of nitrogen metabolism in sediment, and lower functional redundancy of nitrogen metabolism and phosphorus cycle in water column. In this eutrophic river-lake system, functional compositions of the bacterial communities were vulnerable to nutrient perturbations especially in water column. By revealing the metabolism pathways of major functions, the influences of nutrient variables on functional compositions, and functional redundancy, this study can provide insights into the microbial community structures and ecological processes in this river-lake system.

## ACKNOWLEDGEMENTS

We are grateful to the anonymous reviewers for the comments, to Yuhang Zhang and Chenyu Yang for their assistances in the field and laboratory work.

### Funding

This study was supported by the Project of State Key Laboratory of Simulation and Regulation of Water Cycle in River Basin (SKL2018CG02), the National Natural Science Foundation of China (No. 51439007), and the IWHR Research and Development Support Program (WE0145B532017). The funders had no role in study design, data collection and analysis, decision to publish, or preparation of the manuscript.

### Grant Disclosures

The following grant information was disclosed by the authors:
Project of State Key Laboratory of Simulation and Regulation of Water Cycle in River Basin: SKL2018CG02.
National Natural Science Foundation of China: 51439007.
IWHR Research and Development Support Program: WE0145B532017.

### Competing Interests

The authors declare that they have no competing interests.

### Author Contributions

- Ze Ren conceived and designed the experiments, performed the experiments, analyzed the data, prepared figures and/or tables, authored or reviewed drafts of the paper, approved the final draft.

- Xiaodong Qu conceived and designed the experiments, performed the experiments, analyzed the data, prepared figures and/or tables, authored or reviewed drafts of the paper, approved the final draft.
- Wenqi Peng performed the experiments, contributed reagents/materials/analysis tools, approved the final draft.
- Yang Yu performed the experiments, contributed reagents/materials/analysis tools, approved the final draft.
- Min Zhang performed the experiments, contributed reagents/materials/analysis tools, approved the final draft.

### Data Availability

The raw sequence data are available at National Center for Biotechnology Information (PRJNA436872, SRP133903).

### Supplemental Information

Supplemental information for this article can be found online at http://dx.doi.org/10.7717/peerj.7318#supplemental-information.

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
