# Peer review of "Functional properties of bacterial communities in water and sediment of the eutrophic river-lake system of Poyang Lake, China"

_PeerJ, doi:10.7717/peerj.7318_

## Round 0.1 · original submission · Major Revisions

This paper has provided some new information about the compositional and functional difference of bacterial community between sediment and water samples in the river-lake system of Poyang Lake. However, major revisions are needed before the paper can be accepted. The sampling procedure is not clear enough. More environmental data and analyses are required. Also the English needs to be further improved.

Reviewer 1 ·

Basic reporting

The manuscript entitled “Functional properties of bacterial communities in water and sediment of the eutrophic river-lake system of Poyang Lake, China” by Ren et al. discusses microbial functional potential in lake and river sediments and water column of the Poyang Lake, China. Authors offer new insights to the function of microbial communities in eutrophic waters and also explore the functional redundancies of the communities. I particularly enjoyed the breakdown of potential functions for each of the biogeochemical cycles, and the comparison between sediments and water column.

However, there are few parts of the manuscript that need expansion and clarifications.
The manuscript, particularly the introduction and discussion, is difficult to follow due to many grammatical errors and lack of proper punctuation. The standard of English needs to be improved and the paper requires a thorough editing for grammar, punctuation, word choice, etc. The language is too colloquial at times, and reoccurring grammatical mistakes make the manuscript hard to follow.

Experimental design

The manuscript lacks important data and details on sampling procedures. I listed most of the in the comments below. Main concerns include: lack of environmental data and analysis (temp, DO, pH, Chlorophyll, etc.), no information on the status of this eutrophic lake (cyanobacterial blooms? Turbidity? Sampling depths?). Please see detailed comments below and clarify.

Validity of the findings

I wish the authors expanded more on the association between functional redundancy and the vulnerability to environmental perturbations.
I understand the concepts of high and low functional redundancies, but the authors did not expand on these findings and what they mean for the ecosystem and microbial dynamics? What does it mean for biodiversity? Why is this important? This part of the discussion (and conclusions) needs more work.

Additional comments

More specific comments

Line 17: define PICRUSt

63: This first sentence sounds incomplete and confusing

67: Run-on sentence; confusing

103: No specific information on the lake, its tributaries, and sample collection. How deep is it? What’s the average depth of the rivers? How deep were the sediments collected? At what depth were the water samples collected?

110: I appreciate that all these nutrients were collected and analyzed but this manuscript is lacking environmental variables that are likely impacting biogeochemistry and microbial distribution. Providing env. variables like temperature, dissolved oxygen, pH, turbidity, and chlorophyll a concentration would be beneficial and including them in statistical analysis, along with nutrients, could reveal new and important relationships.

124: reference for the primers?

216: grammar; “This study showed”

224: defined OM in sentence 222, no need to do it again

225: “contributed” ?

227: “consistent of” seems incorrect here

240: Is the lake actually eutrophic? Is there any history of cyanobacterial blooms? This is important information that should be addressed in the discussion. Blooms can have a big impact on biogeochemical cycles and microbial communities in water column and sediments (i.e., Taihu).

250: Is this complete denitrification ending at N2? Or just N reductase genes (nas, nor, nir, nos) This is important as denitrification is a stepwise reaction and it can sometimes stop at N2O.

259: This goes along with the previous question. In the presence of oxygen, denitrification can stop at N2O because nos is oxygen sensitive. It is important to clarify here, whether this high potential of denitrification was in fact complete denitrification to N2 gas. Or just high potential of NO3- reductions.

256: This sentence is grammatically incorrect

272: Nitrification is also an important process in the N cycle, especially in the shallow lakes where it can be coupled with denitrification.

274: “can help us understand”

285-286: this sentence is confusing and awkward

295: I don’t see any mention of DNRA, which can be happening in the presence of sulfur as well. Did DNRA play a role in the sediments? Was there a functional potential for it?

307: This sentence is again confusing and grammatically incorrect

·

Basic reporting

Ren Z et al. have reported the distinct compositional and functional difference of bacterial community between sediment and water samples from the river-lake system of Poyang Lake. The English language of this paper is adequate, and the structure conforms to PeerJ standards. I still concerned two points,
1, The nutritional status of Poyang Lake and its river-lake system is described as eutrophic in this paper, but actually it’s periodly eutrophic, and not appreciate to be called as eutrophic.
2, There is no evident difference between river and lake regions from the results of either water or sediment samples, this point is quite strange in such an estuary ecosystem, and I think some interpretation are needed in the discussion.
Besides, literature was well referenced, but the format was quite confusion in the list.
The authors have supplied raw sequencing data, but the sample name was inconsistent and hard to identify.

Experimental design

Although this paper seemed as a re-analysis of previous study by bioinformatics methods, it’s original and significative and worth to report, and within the scope of the journal.
The introduction part is not clear and coherent enough, and also the scientific issue and knowledge gap need carefully stated.
The statistical analysis section should be more carefully described, and supplement some exemplary literatures for the models.
The discussion section only focused on the metabolic function, while the sampling information such as the flow direction, environmental pollution, and river-lake interaction was not discussed.

Validity of the findings

The paper compared the functional redundancy of bacterial community by a linear regressions model, in which the functional composition dataset was based on the predicted metagenomes and their KEGG annotation, while the taxonomic composition was based on the whole sequenced community, and functional composition represented only a subset of the taxonomic composition. The author could try to eliminate the unidentified taxonomic groups, and then assess the function-composition relationship.

Additional comments

I think this study is of great significance for the research of Poyang Lake, and the function prediction give us some specific insight for a microbial community. However, I think the writing is too concise that many important details are overlooked (methods), and the introduction and discussion also should be improved.

Reviewer 3 ·

Basic reporting

The English was satisfactory although there were some typos and confusing sentences throughout. Many statements were not cited and need to be supported. Suggested improvements and specifics are posted below.

Experimental design

Methods need to provide much more details, especially within the statistical analysis section and the analysis of the sequence data.

Validity of the findings

Statistics definitely need to be reported and perhaps more rigorously tested. The importance or significance of the findings are not clear in relationship to the literature that was discussed nor to the greater overall question.

Additional comments

Line 63: this is a redundant sentence of the first in the intro.
Line 69: remove the word “all.”
Line 78: I would recommend this paragraph start with a comment on how there are many different ways to characterize functional capacity and then just discuss PICRUSt because I am left wondering why you wouldn’t use metagenomes over 16S predictions.
Line 97 and line 104: reference source not found ?
Line 103: is there a map somewhere indicating where these samples were collected?
Line 109: clarify how the sediment was homogenized
Line 139: quality?

Statistical Analysis: clarify what exactly was being compared. Abundances or presence absence? Was the data transformed? Etc.
Line 154: I don’t understand this
Line 162: Clarify what significance of nutrient variables means
Line 168: were these abbreviations defined earlier in the text?

There’s no reference to figure 1 in the text.

Line 169-170: report p values in text or reference to figures where this information can be found in text.
Line 201-210: I still don’t understand how you get redundancy information out of linear regression please clarify in methods or results.

Line 216: cite this statement… sediments can heavily influence the water column, are these truly distinct communiites?”
line 231: confusing sentence, please clarify and cite.
Lines 233: A lot of discussion of OM but how does this relate to your results? What about carbon fixation, methane production?
Line 238: I believe this is much larger, I would find a more recent citation.
Line 252: was OC defined earlier in text?
Line 259: Dynamics in environmental parameters in this lake is not discussed in the results and it is unclear if you are getting this info from new data collected in this paper or literature.
Line 276; but if there are high N loads, its probably not going to be occurring. If you think this is not the case, please support it in the discussion. Also support the significance of this more clearly. Nif genes are wide spread in make prokaryotes and archaea so it isn’t that surprising you see a lot of potential from this.

Lines 290 and beyond: need to support these statements with citations.
Line 301: again why is this important? What is the significance of these statements in relation to this system?
Line 317: again, what is the significance of this? What does this mean in the greater picture of lake poyang?

Figure 3. p values need to be reported somewhere. In 3A, it is hard to believe that there are truly significant differences between carbohydrate metabolism considering those error bars. Same with methane production. All the raw statistical data needs to be provided because it seems like B-D is also the case where the error bars are overlapping a lot and don’t really look different. If you are doing a lot of ANOVAs, corrections need to be made for multiple comparisons as well.

Figure 6. Should this just be a table?

Why was there not analyses done with just the 16S data? For example the abundances of 16S genes. I think this would be more convincing than using the PICRUSt information as this is all predicted.

---

## Round 0.2 · accepted · Accept

The manuscript has been improved significantly after the revision. It meets the publication standard.